# Automating Structural Engineering Workflows with Large Language Model Agents

## Abstract

We introduce **MASSE**, the first Multi-Agent System for Structural Engineering, effectively integrating large language model (LLM)-based agents with real-world engineering workflows. Structural engineering is a fundamental yet traditionally stagnant domain, with core workflows remaining largely unchanged for decades despite its substantial economic impact and global market size. Recent advancements in LLMs have significantly enhanced their ability to perform complex reasoning, long-horizon planning, and precise tool utilization—capabilities well aligned with structural engineering tasks such as interpreting design codes, executing load calculations, and verifying structural capacities. We present a proof-of-concept showing that most real-world structural engineering workflows can be fully automated through a training-free LLM-based multi-agent system. MASSE enables immediate deployment in professional environments, and our comprehensive validation on real-world case studies demonstrates that it can reduce expert workload from approximately two hours to mere minutes, while enhancing both reliability and accuracy in practical engineering scenarios.

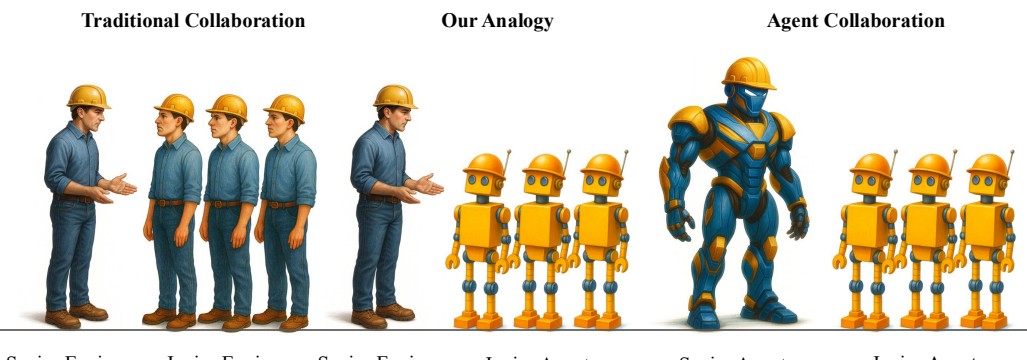

Figure 1: **Analogy of future human–AI collaborations.** Traditional practice relies on long apprenticeships, with senior engineers transferring expertise to junior engineers through mentorship and problem solving. In contrast, LLM-based multi-agent systems instantiate scalable junior engineer agents that inherit workflows, perform specialized tasks, and coordinate under senior engineers' oversight. As these systems evolve toward self-planning and adaptive learning, fully agentic hierarchies—with senior engineer agent directing junior engineer agents—could transform engineering into a continuously improving, highly efficient practice.

## 1 Introduction

Structural engineering, the discipline responsible for designing and building of structures including buildings, bridges, and other infrastructure, has shaped human civilization for millennia—from pyramids and aqueducts to skyscrapers and transportation networks (Saouma, 2010; Addis, 2007). What began as intuition and craftsmanship has become a science grounded in mechanics, materials, and computation (Salvadori, 1990; Allen, 2001). Today, engineers translate ambiguous requirements into models, simulate loading scenarios with specialized software (e.g., OpenSees, SAP2000,

Abaqus/CAE) (McKenna, 2011; Pasticier et al., 2008; Wang & Melly, 2018), and verify compliance against extensive building codes (National Research Council of Canada, 2020; Canadian Institute of Steel Construction, 1992; Reynolds et al., 2007; Stalnaker & Harris, 1997). Yet despite underpinning a multi-trillion-dollar global sector (Barbosa et al., 2017), structural engineering remains among the least digitalized industries, hindered by fragmented workflows, manual knowledge transfer, and coordination bottlenecks (ASCE, 2019; Agarwal et al., 2016; KPMG, 2019). The result is inefficiency, cost overruns, and lost opportunities for sustainability and resilience.

In parallel, large language models (LLMs) have transformed AI. Scaling transformer-based architectures (Vaswani et al., 2017) to billions of parameters yields broad generalization (Bommasani et al., 2021), predictable scaling laws (Kaplan et al., 2020; Hoffmann et al., 2022), and emergent competencies suggestive of general intelligence (Bubeck et al., 2023; Feng et al., 2024). Advances such as chain-of-thought prompting (Wei et al., 2022; Kojima et al., 2022) and instruction tuning (Ouyang et al., 2022) have unlocked strong reasoning, while frontier models—GPT-4o (OpenAI, 2024), GPT-5 (OpenAI, 2025), Qwen-3 (Yang et al., 2025), DeepSeek-R1 (Guo et al., 2025a)—achieve unprecedented performance in math, code, and tool use. The trajectory has shifted toward *agents*: LLM systems that plan, invoke tools, and coordinate tasks (Yao, 2025; Silver & Sutton, 2025). Frameworks such as ReAct (Yao et al., 2023b), Tree of Thoughts (Yao et al., 2023a), Voyager (Wang et al., 2023), and StateFlow (Wu et al., 2024) externalize reasoning into structured workflows, enabling deployment in domains that are procedural, codified, and tool-centric.

Structural engineering tasks, particularly structural design, exemplifies a domain where tasks are verbalizable, procedural, and tool-centric: requirements map to models, load cases to simulations, and code clauses to compliance checks. Single LLMs can aid structural analysis (Liang et al., 2025a), yet their accuracy collapses when tasks demand multi-tool calls or chained subtasks, especially with complex geometries, underscoring the need for a resilient multi-agent framework (see Figure 2). We present **MASSE**, a *multi-agent system* tailored to structural design tasks. MASSE operationalizes professional workflows by assigning specialized LLM agents to distinct roles—*Analyst* (data and code extraction), *Engineer* (modeling and limit-state checks), and *Manager* (coordination and final adequacy decision). These roles are orchestrated through the AutoGen (Wu et al., 2023), supported by structured memory for persistent analysis data. By embedding FEM solvers and code documents directly in the loop, MASSE achieves stable tool-augmented reasoning, formal safety verification, and reduces hours of expert iteration to minutes. Therefore, we hypothesize that such agent-driven workflows can generalize beyond structural engineering to other domains where tasks are verbalizable, procedural, and tool-mediated (see Figure 1). Our contributions are:

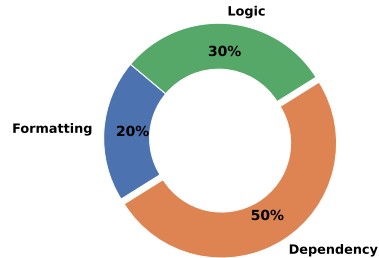

Figure 2: **Failure Modes of Single-Agent.** We evaluate the same structural engineering problem over 10 trials, comparing single-agent and multi-agent setups. The single-agent failed in all 10 trials, with error distributed as shown in the figure. In contrast, our multi-agent framework succeeded in every trial. See details in Appendix A.

- We pioneer an LLM-based multi-agent system that mirrors end-to-end structural design workflows with explicit safety and verification loops.

- We introduce a new dataset and case studies grounded in real-world problems, demonstrating automation of complex, tool-mediated tasks.

- We demonstrate substantial reductions in expert time while preserving accuracy, supporting the broader thesis that verbalizable, tool-centered professional workflows can be automated at scale.

## 2 RELATED WORK

### 2.1 LLMS IN CIVIL ENGINEERING APPLICATIONS

LLMs have been extensively applied in Civil Engineering and related domains (Xie et al., 2025). For instance, natural language–to–code frameworks have advanced structural optimization (Qin et al., 2024), prompting and in-context learning have enhanced structural analysis pipelines (Liang et al.,

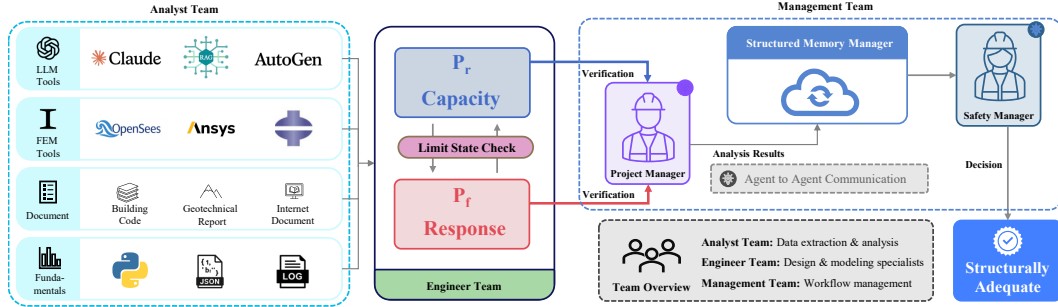

Figure 3: *MASSE* Overall Framework. The **Analyst Team** combines four layers of tools (LLMs, FEM solvers, engineering documents, and fundamentals) to enable multi-agent collaboration. The **Engineer Team** performs limit state verification (Response<Capacity), while the **Project Manager** coordinates workflows and the **Structural Memory** stores analysis data. The **Safety Manager** conducts the final adequacy check. Three specialized teams are shown: **Analyst** (data extraction and analysis), **Engineer** (design and verification), and **Management** (coordination and decision-making).

2025a; Avila et al., 2025), and multi-agent router systems have improved foundation design automation (Youwai et al., 2025). LLM-driven frameworks further advanced code-compliant reinforced concrete design (Chen & Bao, 2025), LLM-based agents enabled reliable and robust automation of structural analysis (Liu et al., 2025), and LLM-based BIM authoring systems streamlined modeling tasks (Du et al., 2025; Deng et al., 2025). Multi-agent collaboration has also been applied to Ultra-High Performance Concrete (UHPC) design, enabling knowledge-guided and data-driven material development (Guo et al., 2025b). Vision-language systems contributed to safety and compliance through ergonomic risk assessment (Fan et al., 2024), structured hazard detection (Adil et al., 2025), and RAG-based engineering code consultation fine-tuned on NBCC data (Joffe et al., 2025; Aqib et al., 2025). DrafterBench (Li et al., 2025a) benchmarks such LLM-driven automation in civil engineering tasks.

## 2.2 MULTI-AGENT SYSTEMS WITH LLMS

Recent advancements in multi-agent systems (MAS) have demonstrated the potential of LLM-powered agents to collaborate effectively across domains such as chemistry (Tang et al., 2025), healthcare (Schmidgall et al., 2024), and finance (Xiao et al., 2024), where specialized roles improve domain-specific performance. In software engineering, frameworks like ChatDev (Qian et al., 2023), HyperAgent (Phan et al., 2024), MetaGPT (Hong et al., 2023), and Magentic-One (Fourney et al., 2024) coordinate agents for design, coding, and testing, while general-purpose infrastructures such as AutoGen (Wu et al., 2023) and AppWorld (Trivedi et al., 2024) provide flexible environments for orchestration and benchmarking. At the same time, researchers have begun systematically analyzing MAS limitations, highlighting failure modes such as misaligned objectives and verification bottlenecks (Cemri et al., 2025; Microsoft Corporation, 2025), and stressing the need for rigorous benchmarking practices that jointly optimize accuracy and efficiency (Kapoor et al., 2024). Complementary to these system-level advances, efforts to enhance the efficiency and interpretability of LLMs themselves include model compression (Lin et al., 2024; Kumar et al., 2024; Liang et al., 2025b; Shen et al., 2025d), inference acceleration techniques (Shen et al., 2025b;c;a; Jiang et al., 2025; Shao et al., 2025), and theoretical analyses of their representational power (Chen et al., 2025a;b; Liang et al., 2025c).

## 3 SYSTEM ROLE DESIGN

In this section, we introduce the specific agent role design of our MASSE framework. We clearly define roles and specific goals for LLM agents, enabling complex structural engineering tasks to be efficiently segmented into smaller, manageable components. Structural engineering is inherently multifaceted, requiring diverse inputs, specialized knowledge, and coordinated expertise, such that professional practice relies heavily on multidisciplinary teams to collaboratively address complex, high-stakes decisions (Wolff & Luckett, 2013; Connor & Faraji, 2016; Zhang et al., 2021; Awomolo

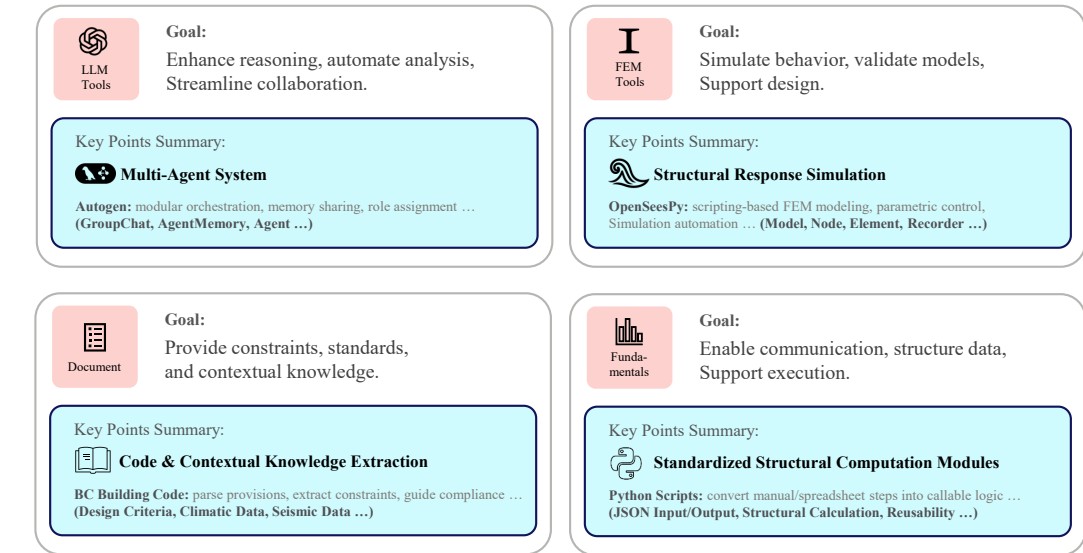

Figure 4: MASSE Analyst Team with Tool Integration

et al., 2019; Chen & Liew, 2002; Rocha et al., 2022), a workflow that LLM agents are well positioned to replicate and automate.

## 3.1 TRADITIONAL WORKFLOW

In conventional structural engineering practice, the design process is carried out through a sequence of tightly coupled steps, beginning with consulting building codes to obtain site-specific seismic parameters or climatic data. Engineers then collect geometric information from site measurements or architectural drawings and determine applied loads—including dead loads, live loads, and other types of loads—using code-based occupancy provisions and spreadsheets or rule-based calculations. With these inputs, they evaluate the load-bearing capacities of structural members from section dimensions and material properties, and use either hand calculations or finite element models to simulate structural responses under multiple load combinations. If requirements are not met, the workflow must be repeated with adjusted assumptions or revised sections. For specialized problems such as warehouse racking systems, this process may take several hours per rack, and with multiple racks per site, the workload becomes extensive. Overall, the traditional workflow is time-consuming, error-prone, and difficult to scale, motivating the need for automation frameworks such as MASSE.

## 3.2 OUR SYSTEM

Reflecting this practical organizational model, MASSE (Figure 3) introduces three distinct agent teams within a simulated structural engineering consultancy environment: Analyst Team, Engineer Team and Management Team. For comprehensive descriptions of the system instructions assigned to each agent, see Appendix B. Each agent is assigned a unique role, goal, and set of constraints, and is further equipped with predefined contexts and specialized tools aligned with these responsibilities. We will then introduce how MASSE organizes these agent roles into the following structured teams. More detailed descriptions could be found in Appendix C.

## 3.3 ANALYST TEAM

The Analyst Team (Figure 4) automates the preparation of structural engineering data by coordinating specialized agents that extract loading conditions, retrieve information from engineering documentation, execute load determination using rule-based methods, and generate structural models. At a high level, this team transforms unstructured project information and regulatory data into standardized engineering inputs, ensuring that subsequent model, design and verification tasks can be carried out with consistency, efficiency, and scalability.

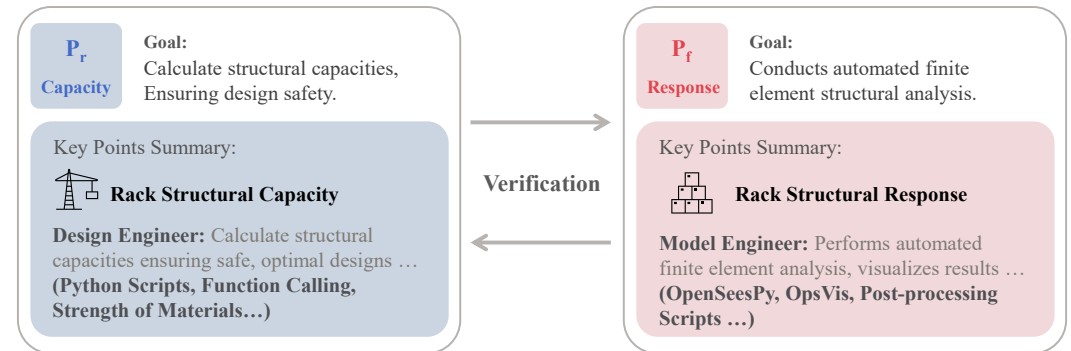

Figure 5: MASSE Engineer Team: Analyze, Design and Verify

## 3.4 ENGINEER TEAM

The Engineer Team (Figure 5) operationalizes the data prepared by the Analyst Team by conducting structural analysis, structural design, and adequacy verification. In broad terms, this team integrates automated simulations and capacity checks to evaluate structural integrity under prescribed loading conditions, enabling systematic, tool-driven assessment of design safety and structural performance.

## 3.5 MANAGEMENT TEAM

The Management Team oversees and coordinates the overall MASSE workflow, transforming analytical outputs from the Analyst Team and Engineer Team into authoritative engineering decisions. Broadly speaking, this team manages task allocation, integrates intermediate results, and issues final structural safety conclusions that guide the entire system. As illustrated in Figure 6, the Safety Manager plays a central role within this process by delivering the ultimate adequacy verdict, ensuring that all decisions are consistent with professional safety standards.

## 4 AGENTIC WORKFLOW DETAILS

### 4.1 BACKBONE MODELS

In our system, agents are configured to be initialized with powerful LLMs (e.g., GPT-4o, Claude 3.5 Sonnet), or reasoning models (e.g., o4-mini). We also use Embedding models (e.g., text-embedding-3) to support retrieval-augmented generation for accessing building codes and technical documents. Temperatures are set to 0 for GPT-4o, Claude 3.5 Sonnet, and GPT-3.5-turbo, and 1 for o4-mini.

### 4.2 COMMUNICATION PROTOCOL

MASSE prioritizes structured-format communication over verbose natural language exchanges. Natural language reliance often degrades performance in structural engineering tasks that require extended planning horizons, as verbose dialogue can overflow context windows and obscure key details. To address these issues, MASSE adopts structured communication paradigms, where each agent's operational state is defined through formalized protocols, such as JSON-based input–output schemas and turn-level state tracking mechanisms. This ensures concise, verifiable outputs while reducing redundancy and distortion.

### 4.3 AGENT INTERACTIONS

We design agent interactions around structured artifacts that encode analytical results, ensuring traceability and systematic integration. We enforce predefined protocols with input–output formats, schemas, and validation rules to eliminate ambiguity and enable orchestration. In this pipeline, we let analysts convert noisy problem descriptions and unstructured documents into structured memory, engineers transform them into deterministic outputs such as finite element analysis results and capacity

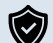
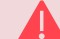

**Goal:**
Oversees risk assessment, code compliance, and final safety assurance

Safety Manager

**Key Points Summary:**

**Final Safety Assessment**

**Safety Verdict:** Confirms adequacy, flags risks, enforces compliance…
**(Regulatory Checks, Failure Traceability, Termination Logic …)**

**Decisions**

⚠️ **STRUCTURALLY INADEQUATE**

**Reasoning:**
The 6-inch step beam's capacity is insufficient to support the uniform load from cargo weight.

**Recommendation:**
Reduce cargo weight or replace the beam with a stronger section.

Figure 6: MASSE's Safety Manager's Decision-Making Process

checks, and management roles validate outputs against codes before authorizing final decisions. When inconsistencies occur, we allow brief natural language exchanges to recover missing information, but all outcomes are distilled back into structured form to preserve accountability. We log every exchange for auditing and debugging, and the overall framework mirrors professional engineering workflows where domain-specific expertise is coordinated through standardized communication. A detailed trajectory of a racking system case is provided in Appendix D.

## 5 EXPERIMENTS

### 5.1 SIMULATION SETUP

We evaluate MASSE in a racking system design scenario (Castiglioni et al., 2016), a representative task in structural engineering where engineers must determine allowable loads and ensure compliance with seismic safety requirements. This setup integrates core engineering steps—data retrieval, load transformation, structural modeling, analysis, and design—directly aligning with the functionality of MASSE agents. Full details of the simulation configuration are provided in Appendix E.1.

### 5.2 DATASET

We constructed a domain-specific dataset consisting of one hundred different levels of difficulty, each paired with validated ground-truth solutions, enabling robust evaluation reflective of real-world engineering challenges. To comply with privacy constraints, the released dataset is reorganized but faithfully mirrors the original cases, ensuring reproducibility of system performance. Each sample contains a natural language problem description, intermediate reasoning steps, and final results used as evaluation criteria. Notably, loading reports are mandated and must be certified by structural engineers in earthquake-prone regions such as British Columbia and California, underscoring the practical significance of this scenario. Our modular design allows straightforward adaptation to many structural engineering tasks, making them broadly applicable as plug-and-play components. An illustrative racking system problem and its structural scheme are provided in Appendix F.

### 5.3 EVALUATION METRICS

To benchmark MASSE against real-world structural engineering tasks, we design novel evaluation metrics aligned with the core agent roles: Structural Analysis Agent Benchmark (SAAB), Structural Design Agent Benchmark (SDAB), Loading Agent Benchmark (LAB), and the holistic Multi-Agent Structural Engineering Benchmark (MASEB). Each benchmark evaluates the ability to convert natural language inputs into structured analyses, tool executions, and safety decisions, while MASEB additionally incorporates system cost and runtime to balance efficiency and performance. To ensure these benchmarks reflect professional practice, we base our evaluation on data reorganized from real-world projects in British Columbia, Canada. This design allows the metrics to capture both

technical accuracy and the fidelity with which agents replicate the workflow of a consulting firm. Detailed rubrics, task specifications, and evaluation protocols are provided in Appendix E.2.

# 6    RESULTS AND ANALYSIS

## 6.1    MAIN RESULTS

### 6.1.1    PERFORMANCE COMPARISON

Table 1 summarizes MASSE performance across four benchmarks under different LLM backends. Bold numbers denote the best model on each benchmark, while underlined numbers indicate the second best. Each problem was evaluated over ten independent trials, and a total of one hundred traces were collected to assess the overall system performance. Among language models, *Claude 3.5 Sonnet* delivers the strongest overall consistency, leading in SDAB (89.2) and remaining competitive elsewhere, while *GPT-4o* achieves the highest LAB score (98.1) but trails slightly in SAAB and MASEB. By contrast, *GPT-3.5-turbo* underperforms (e.g., 64.6 in SDAB, 67.7 in MASEB).

Table 1: Performance Comparison of MASSE under Different LLMs. The best score per benchmark is in bold, the second-best is underlined. Avg. is reported across four benchmarks.

| Models | SAAB | SDAB | LAB | MASEB | Avg. |
|---|---|---|---|---|---|
| **Language Models** | | | | | |
| *Claude 3.5 Sonnet* | 87.0 | 89.2 | 94.8 | 85.6 | 89.2 |
| *GPT-3.5-turbo* | 71.4 | 64.6 | 90.5 | 67.7 | 73.6 |
| *GPT-4o* | 85.6 | 87.4 | **98.1** | 82.7 | 88.5 |
| **Reasoning Models** | | | | | |
| *o4-mini* | **96.6** | **91.4** | 93.8 | **94.7** | **94.1** |

Reasoning model *o4-mini* outperforms across three of four benchmarks, with peak results in SAAB (96.6), SDAB (91.4), and MASEB (94.7), and only narrowly behind GPT-4o in LAB (93.8 vs. 98.1). This balance highlights the intelligence and robustness of reasoning models, showing that reasoning yields higher reliability across heterogeneous tasks. Overall, the results indicate that (1) reasoning models like *o4-mini* provide the most stable and generalizable performance, and (2) among standard LLMs, large-scale models such as *Claude 3.5 Sonnet* and *GPT-4o* remain essential for high-fidelity structural engineering workflows.

### 6.1.2    COST ANALYSIS

Figure 7a illustrates a clear performance–efficiency trade-off. *o4-mini* achieves the best overall score but with the heaviest computational cost—both in runtime and token consumption. *Claude 3.5 Sonnet* performs slightly worse while also incurring substantial token usage and longer runs. *GPT-4o* offers a balanced middle ground, delivering strong accuracy at a moderate cost. In contrast, *GPT-3.5-turbo* is the most economical and fastest, though its performance lags behind the others. These patterns suggest that MASSE benefits from stronger backbones when quality is paramount, but real-world deployment must balance performance with latency and budget.

### 6.1.3    RUNTIME ANALYSIS

In this experiment, we vary the maximum number of agent-to-agent communication rounds, which is set to 1 to 4. Allowing more rounds of communication enables agents to exchange additional information, identify and fill in missing details, refine intermediate reasoning steps, and apply self-correction before producing final outputs. The evaluation is performed on ten representative problems selected from MASEB. To mitigate randomness in model behavior and ensure robust measurement, each problem instance is independently repeated ten times.

The runtime analysis (Figure 7b) demonstrates the fundamental trade-off in multi-agent coordination. As the number of agent-to-agent communication rounds increases from one to four, runtime rises steadily—from about 20 seconds at a single round to nearly 70 seconds at four rounds—reflecting the

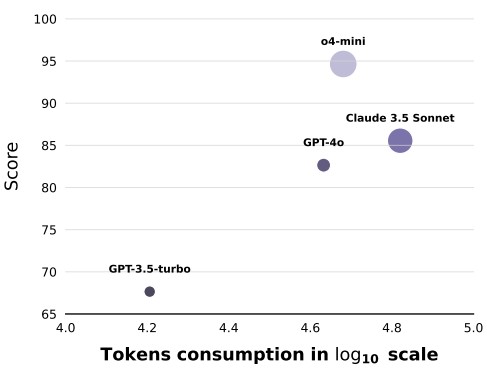

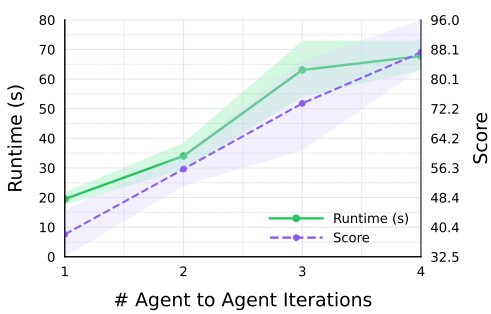

(a) **Cost analysis.** Bubble size denotes average run-time count per method; y-axis shows average score across four datasets.

(b) **Runtime.** X-axis: agent communication rounds; left y-axis: total runtime; right y-axis: test performance. The shaded regions indicate the 95% CI computed via bootstrap.

Figure 7: Results Analysis: (a) cost–performance trade-off and (b) runtime–performance relationship.

additional overhead of iterative dialogue. However, this increase in computational effort correlates with consistent improvements in system score, which grows from below 40 to nearly 90. These results highlight a key feature of multi-agent systems: while deeper rounds of communication impose latency, they also enable richer reasoning and more reliable outputs.

## 6.2 HUMAN EVALUATION

To empirically evaluate MASSE's efficiency, we conducted a comparative study with 11 experienced structural engineers, each independently completing a standardized racking system design task. The conventional workflow—dependent on fragmented software, manual data retrieval, repetitive entry, and interpretive effort—required an average of 132 minutes per task. By contrast, MASSE, powered by GPT-4o, automated these processes and reduced the average completion time to approximately two minutes, a reduction of over 98%. This result demonstrates a marked improvement in both productivity and operational reliability within professional practice. In consulting environments, executing tasks at a fraction of the traditional time and cost (on the order of 1/100) is likely to enhance profit margins, streamline organizational workflows, and redefine competitive positioning. More broadly, firms that strategically adopt and effectively integrate AI systems such as MASSE may not only secure substantial advantages in the engineering sector, but also be positioned to shape new paradigms of practice, potentially influencing how design, safety, and efficiency are balanced across the built environment.

## 6.3 ABLATION STUDY

Table 2: Ablation study on agent components: agent memory $M$ and JSON format $J$.

| Agent Component | SAAB | SADB | LAB | MASEB | Avg. |
|---|---|---|---|---|---|
| None | 52.5 | 65.9 | 81.6 | 47.1 | 61.8 |
| +$M$ | 55.9 | 69.8 | 89.6 | 50.6 | 66.5 |
| +$J$ | 56.6 | 67.2 | 84.1 | 50.1 | 64.5 |
| +$M, J$ | **85.6** | **87.4** | **98.1** | **82.7** | **88.5** |

Agent memory and structured input–output are increasingly recognized as common practices and active research directions in the design of multi-agent systems (Xiong et al., 2025; Li et al., 2025b; Fan et al., 2025). To assess the effect of individual agent components, we performed an ablation study across four benchmarks (Table 2). The baseline configuration without memory or structured I/O yields the lowest scores in all settings. Incorporating multi-agent memory ($M$) improves performance by preserving intermediate reasoning traces across turns, with notable gains in LAB ($81.6 \rightarrow 89.6$).

Similarly, enforcing JSON-based input–output constraints ($J$) provides consistent benefits by reducing ambiguity in communication, especially in SAAB and SADB. The combination of both components ($+M, J$) achieves the strongest results overall, setting the best score on every benchmark. These findings indicate that memory and structured I/O constraints are crucial: memory enhances contextual continuity, while JSON formalization enforces precise data exchange.

## 7 DISCUSSION

**Transparency.** Reliability and accountability are prerequisites for engineering adoption. MASSE is therefore designed with verifiable processes rather than opaque autonomy. Every artifact is logged with explicit reasoning traces, giving practitioners immediate access to the system's decision path and enabling systematic validation. Deterministic outputs further support reproducible quality checks, while localized error handling ensures that small mistakes do not propagate into larger failures. By structuring agents to follow well-defined workflows, MASSE effectively plays the role of a junior engineer whose steps are fully auditable—providing transparency and traceability without conceding control to uncontrolled autonomy.

**Safety.** Because structural engineering involves high-stakes decisions, ultimate responsibility must remain with senior practitioners; even small misjudgments can produce catastrophic outcomes. MASSE is designed to augment, not replace, this expertise by compressing the end-to-end review process from roughly two hours to only a few minutes (Section 6.2). This acceleration allows experts to evaluate a wider range of design alternatives and stress scenarios within the same timeframe, thereby improving robustness and resilience of built systems. Equally important, MASSE produces structured intermediate outputs that mirror professional practice, akin to junior engineers drafting reports for senior verification. As AI systems continue to improve, such workflows may increasingly reduce reliance on traditional junior roles, reallocating human attention to the oversight and judgment that matter most for public safety.

**Real-World Impact.** The efficiency gains of MASSE highlight how domain-specific multi-agent systems can convert the raw capability of frontier LLMs into transformative productivity advantages. Reducing structural design cycles from hours to minutes (Section 6.2) does more than boost users' competitiveness—it has the potential to alter the economics of engineering services by lowering costs, increasing throughput, and enabling firms to take on larger, more complex projects. Freed from routine procedures, engineers can concentrate on creativity, innovation, and safety-critical deliberation. At scale, such reallocation of effort has implications well beyond structural engineering: knowledge-intensive sectors such as architecture, finance, and healthcare could be reorganized around agentic AI collaborators, yielding faster design cycles, more equitable access to expertise, and accelerated progress on global challenges from sustainable infrastructure to resilient urban development. MASSE thus exemplifies how carefully engineered AI systems can bridge the gap between technical advances in LLM orchestration and tangible societal benefit.

## 8 CONCLUSION

We introduced **MASSE**, a multi-agent framework designed specifically for structural engineering, integrating LLM-based agents into end-to-end professional workflows. By coordinating role-specialized agents, MASSE is able to parse technical standards, perform domain-specific computations, and interact seamlessly with engineering scripts and resources. Its structured communication backbone and iterative reasoning cycle allow the system to achieve robust, high-quality outcomes. Safety verification is embedded through a dedicated management role, and the modular design enables flexible adaptation across a wide range of engineering tasks. Our evaluation on real-world case studies shows that MASSE can reduce expert workload by approximately 98%, cutting the required time from hours to minutes while maintaining both reliability and accuracy. This proof-of-concept demonstrates the feasibility of automating structural engineering workflows without task-specific training. Looking ahead, future work will emphasize deployment in consulting environments, refinement of agent specialization, and integration of real-time feedback to support adaptive self-improvement.

ETHIC STATEMENT

**Research and Educational Purpose Only.** The multi-agent system, methods, code, and data described in this paper are developed solely for academic research and educational use. They are not intended, nor should they be relied upon, for direct application in real-world engineering design, construction, or deployment. The authors, their institutions, and any collaborators explicitly disclaim all liability for any consequences, including but not limited to structural deficiencies, under-design, damages, or failures, arising from the use, misuse, or adaptation of the methods, parameters, or formulas presented herein. Any attempt to implement or deploy the system in practice is done entirely at the user's own risk and responsibility.

**Human Evaluation.** All human evaluation conducted in this study was performed exclusively for the purpose of benchmarking and comparative analysis. Data generated by participants is kept strictly confidential, used only within the scope of this research, and will not be applied to any commercial, industrial, or real-world engineering activities.

**Privacy and Data Protection.** The dataset employed in this work is derived from production records within structural engineering consultancy practice but has been anonymized, cleaned, and used strictly for academic research. No identifying information, proprietary designs, or sensitive client data are disclosed. All outputs generated by the multi-agent system remain confined to research purposes and will not be released, applied, or repurposed for operational engineering, deployment, or commercial use. This research adheres to applicable data privacy principles and safeguards the rights and interests of all parties involved.

LLMS USAGE STATEMENT

We disclose that LLMs were employed exclusively as auxiliary tools, limited to (i) refining the exposition of the manuscript for clarity and conciseness, and (ii) generating preliminary schematic components used in visualizations to illustrate methodological pipelines.

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

# Appendix

## A  SINGLE- VS TWO-AGENT EXPERIMENT

Structural engineering workflow usually includes long-horizon tasks. Recent study proves that, without chain-of-thought, non-thinking models struggle to chain even two steps in a single turn (Sinha et al., 2025). An obvious solution is to split these steps across two models to execute sequentially. Therefore, we design an experiment to demonstrate that a multi-agent system is necessary for long-horizon tasks in structural engineering. This appendix reports a controlled comparison between two agent systems for generating OpenSeesPy code from natural language problem descriptions (PD): (1) a two-agent collaborative system (AgentA + AgentB) and (2) a single-agent end-to-end system (AgentAB). The primary goal of this experiment is to evaluate the execution success rate.

For the two-agent system, AgentB extracts parameters from PD into JSON and computes section properties, which are then passed to AgentA for generating complete OpenSeesPy scripts. In contrast, the single-agent system (AgentAB) directly converts PD into executable scripts in one step. To ensure fairness, both systems employ the same model (GPT-4o) under identical environments (`temperature=0.0`). The PD itself is formulated as a two-dimensional structural analysis task involving a two-step procedure. In the first step, material and geometric properties are extracted from the problem description, and a Python script is invoked to compute intermediate parameters such as the moment of inertia and cross-sectional area. In the second step, a finite element model is constructed using both the original parameters from the PD and the derived intermediate values. Outputs were standardized to executable Python scripts performing model assembly, load application, finite element analysis via `openseespy`, visualization via `opsvis`, and printing displacements, reactions, and internal forces.

For evaluation, we focus on execution success rates and systematically characterize the associated failure modes. The results clearly indicate that the two-agent system consistently outperformed the single-agent system in execution success rates. Specifically, the single-agent system failed in all ten trials (see Figure 2), primarily due to cascading errors in parsing and modeling logic. By contrast, the two-agent system achieved robust parameter extraction and stable execution across all cases, albeit with slightly higher latency. In summary, the findings highlight the importance of decomposition and specialization. The multi-agent system introduced modest overhead but ensured higher correctness and execution reliability.

## B  MASSE WORKFLOW: ROLE SPECIFICATION

This appendix outlines the organizational structure of agent roles within MASSE, detailing how specialized teams coordinate to address real-world structural engineering problems. The framework is anchored around three primary groups—the *Analyst Team*, the *Engineer Team*, and the *Management Team*—each responsible for distinct stages of the problem-solving pipeline.

The distribution of responsibilities across these roles draws inspiration from established practices in engineering consultancy, where complex projects are decomposed into well-defined tasks that are either managed individually by an engineer or collaboratively across teams. MASSE reflects this dual perspective: on the one hand, it mirrors the multi-tasking strategies of a single engineer navigating coupled subtasks, while on the other, it replicates the collaborative dynamics observed in professional firms. The system's design principle is straightforward—once a large, long-horizon problem is partitioned into precise subtasks that can be verbalized with clarity, each subtask becomes tractable for a single LLM-based agent. This transformation allows the originally complex workflow to be executed seamlessly through a multi-agent system. The sections that follow present detailed specifications of these agent roles and demonstrate how their interaction yields transparent, resilient, and domain-grounded engineering outcomes.

| Analyst Team Instruction Prompts |
| --- |
| **1. Loading Analyst** |

```
system_message = """
Extract building information from racking system description and return as JSON:

"location": "city, province/state",
"building_type": "racking_system",
"floor_elevations_ft": [list of elevations in feet],
"loads_lbs": [list of loads in pounds],
"dimensions":
"width_ft": number,
"height_ft": number,
"beam_length_ft": number
,
"structural_info": "column and brace specifications"

Extract exact numerical values from the description.
"""
```

### 2. Seismic Analyst

```
system_message = "You extract seismic data from building codes. Return only JSON. No
explanations."
user_message = f"""Extract seismic parameters for location from the document below.
Document:
context
Find the exact numerical values for location and return ONLY this JSON:

"Sa_02": <number>,
"Sa_05": <number>,
"Sa_10": <number>,
"Sa_20": <number>,
"PGA": <number>,
"PGV": <number>

Rules:
- Return ONLY the JSON object
- Use actual numerical values from the document
- If location is not found, return: "error": "City not found"
- No explanations, no text outside JSON"""
```

### 3. Dynamic Analyst

```
system_message = """You are a Dynamic Analyst.
Get data from memory, then call
calculate_seismic_loads(floor_elevations_ft, loads_lbs, seismic_parameters)."""
```

### 4. Structural Analyst

```
system_message = """You are a StructuralAnalyst.

** CRITICAL EXECUTION ORDER **
STEP 1: Create load application nodes at EXACT required elevations FIRST – these are
MANDATORY and cannot be omitted
STEP 2: Create all brace connection nodes at their exact coordinates
STEP 3: Create column nodes for beam-column elements
STEP 4: Verify all required nodes exist before creating elements

Provide output strictly as JSON matching the following format (same keys, types,
order). Return only the JSON.
CRITICAL INSTRUCTIONS:
1. Generate a structural model based on the geometry description provided.
2. **ALL COORDINATES MUST BE IN FEET** – Keep coordinates in feet, do NOT convert to
inches.
3. Forces are in kip, stiffness is in kip/in².
4. **LOAD NODES ARE MANDATORY** – You MUST create load application nodes BEFORE any
other nodes.
5. **CREATE ALL BRACES EXACTLY AS SPECIFIED** – You MUST create truss elements using
the EXACT coordinates provided in the description.
6. Use sequential node IDs starting from 1.
7. AVOID duplicate nodes – merge nodes with same coordinates to prevent zero-length
elements.
8. **BRACE COORDINATES ARE MANDATORY** – For each coordinate pair (x1,y1)->(x2,y2) in
the description, create nodes at those EXACT coordinates and connect them with a truss
element.
9. **DO NOT MODIFY BRACE COORDINATES** – Use the coordinates exactly as written, do not
change them or infer different positions.
10. **VERIFY**: Count your truss elements before finishing – must match the required
count exactly.
11. **UNITS**: Keep all coordinates in feet as provided in the description.
12. **ELEMENT TYPE CONSISTENCY** – Model braces strictly as 2-node axial-only `truss`
elements; use beam/column elements (e.g., `elasticBeamColumn`) for framing members
where bending is required, keeping section units consistent with stiffness units.
```

```
13. **BOUNDARY CONDITIONS FIRST CLASS** - Explicitly define base restraints (e.g., fix
supports with (ux=uy=rz=0) for 2D) before applying any loads; all other DOFs remain
free unless specified.
14. **LOAD PATTERN DISCIPLINE** - Create named load patterns (e.g., Gravity, SeismicX)
and apply forces **only** at the designated LOAD NODES; do not assign moments to
truss-only nodes or DOFs not supported by the element type.
15. **CONSISTENT ORIENTATION** - When defining each element, list the i-end as the node
with smaller x (or smaller y if x ties) to maintain consistent element orientation for
checks, outputs, and post-processing.
16. **POST-BUILD VALIDATION** - After assembly, verify: no zero-length elements, every
mandatory brace is present, each non-load node participates in at least one element,
and reported counts of nodes/elements/load nodes match the expected totals; abort and
correct if any check fails.
17. **DOCUMENT INPUTS CLEARLY** - Record all geometry, section properties, boundary
conditions, and load definitions in comments or separate metadata files to ensure the
structural model is fully reproducible and auditable.
Required JSON format:

"units":
"length": "ft (feet)",
"force": "kip",
"stiffness": "kip/in^2"
,
"materials":
"E": 29000.0
,
"sections":
"column":  "A": 0.705, "I": 1.144 ,
"brace":   "A": 0.162
,
"nodes": [
// Create nodes for all structural geometry points AND load application points
// Example:  "id": 1, "x": 0.0, "y": 0.0
],
"elements": [
// Create beam-column and truss elements based on description
// Include ALL column elements and ALL brace elements as specified
// For braces: Use EXACT coordinates from description, do not modify or infer positions
// Example:  "id": 1, "type": "elasticBeamColumn", "nodes": [1, 2], "section":
"column", "matTag": 1, "transfTag": 1
// Example:  "id": 2, "type": "truss", "nodes": [3, 4], "section": "brace", "matTag": 1
],
"supports": [
// Fixed supports at base nodes
// Example:  "node": 1, "fixity": [1, 1, 1]
],
"loads": [
// Will be populated automatically based on load application nodes
]
"""
```

Figure 8: Instruction prompts for the *Analyst Team*, including Loading, Seismic, Dynamic, and Structural Analysts.

---

**Engineer Team Instruction Prompts**

**1. Design Engineer**

```
system_message = """You are a Design Engineer. Use run_complete_opensees_analysis()
with structural model from memory. One function call completes entire analysis."""
```

**2. Model Engineer**

```
system_prompt_base = """You are a model engineer.
Use run_complete_opensees_analysis() with structural model from memory.
One function call completes entire analysis."""
```

**3. Verification Engineer**

```
system_message = """You are a VerificationEngineer. Use get_analysis_context() to get
all data, then verify_structural_safety(capacities, demands)."""
```

Figure 9: Instruction prompts for the *Engineer Team*: including Design, Model and Verification Engineers.

---

**Management Team Instruction Prompts**

**1. Project Manager**

```
system_message = """
You are a structural engineer. Decompose the racking system problem into specific
inputs.

Extract and return JSON with:

"SDA_input": "Section design: [extract column and brace specifications]",
"LA_input": "Loading analysis: [extract location, loads, heights, dimensions]",
"SAA_input": "Structural analysis: [extract geometry, supports, elements - MUST
preserve ALL brace coordinates exactly as given]",
"number_of_bays": [extract number],
"number_of_pallets": [extract number per beam]

CRITICAL: For SAA_input, you MUST preserve ALL detailed brace coordinates exactly as
they appear in the original description. Do NOT simplify or summarize brace connections
- copy them verbatim with all coordinate pairs.

Focus on extracting exact numerical values and specifications from the description.
"""
```

**2. Safety Manager**

```
system_message = """You are a SafetyManager. Use get_analysis_context() to get all
data. Provide final assessment: "FINAL RESULT: STRUCTURALLY ADEQUATE" or "FINAL RESULT:
STRUCTURALLY INADEQUATE"."""
```

Figure 10: Instruction prompts for the *Management Team*: (1) Project Manager decomposes racking problems into structured inputs for design, loading, and analysis while preserving brace coordinates; (2) Safety Manager performs final adequacy check and outputs the ultimate structural decision.

## C  MASSE COMPONENT DETAILS

This appendix provides detailed descriptions of all core components within the MASSE framework, expanding on the high-level summaries presented in the main text. Each subsection outlines the specific roles, methodologies, and implementation strategies of the corresponding teams and agents, ensuring reproducibility and technical clarity beyond the main paper's length constraints.

### C.1  ANALYST TEAM

The Analyst Team consists of specialized agents tasked with extracting and analyzing critical engineering parameters essential for structural analysis and design. Each agent addresses a distinct aspect of data processing, ensuring comprehensive and automated preparation necessary for subsequent structural analysis and design phases.

- **Loading Analyst**: Specializes in interpreting detailed structural descriptions, converting them into structured engineering data. Leveraging natural language processing and general-purpose LLMs, the Loading Analyst systematically extracts essential building attributes such as floor elevations, applied loads, geometric dimensions, material specifications, and specific engineering criteria. The extracted information is structured into JSON schemas with standardized units, facilitating efficient downstream analyses.

- **Seismic Analyst**: Dedicated to retrieving and compiling seismic design parameters from authoritative databases and regulatory documents. Utilizing Retrieval-Augmented Generation (RAG) alongside advanced semantic search and vector-based retrieval methods such as FAISS indexing, the Seismic Analyst efficiently locates and extracts key seismic data, including spectral acceleration values, peak ground acceleration, and peak ground velocity. The agent employs PDF parsing, embedding models for semantic accuracy, and structured data storage, ensuring precise geographic-specific retrieval and rapid information availability.

- **Dynamic Analyst**: Responsible for calculating seismic loads by accurately integrating structural characteristics with seismic parameters. This agent employs established structural dynamics

methods, including the equivalent lateral force procedure in line with current regulatory codes. The Dynamic Analyst systematically computes base shear forces, lateral load distributions, height-dependent amplification factors, and detailed seismic demands at the story level. Computations are rigorously validated using numerical methods provided by mathematical libraries such as `NumPy`, ensuring adherence to regulatory standards.

- **Structural Analyst**: Acts as an advanced structural model generation specialist. Utilizing AutoGen's `ConversableAgent` framework integrated with general-purpose LLMs, this agent transforms high-level engineering descriptions into detailed, JSON-formatted finite element models compatible with `OpenSeesPy`. Its primary function, `generate_structural_model(description)`, processes input from `StructuralMemoryManager` using prompts, validating critical load nodes for accurate force applications at specified elevations. The agent produces comprehensive structural models detailing geometric coordinates, element connectivity, material properties, section properties for structural elements (columns and braces), boundary conditions, and load vectors, integrating calculated seismic loads. Robust error handling includes JSON parsing and regular expression-based content cleaning to ensure uninterrupted analysis. The Structural Analyst ensures structural engineering compliance, computational efficiency, and effective cross-agent communication through memory management tools.

Collectively, the Analyst Team synthesizes structured engineering data from various sources, including detailed textual descriptions and regulatory documentation. Their coordinated efforts provide comprehensive, reliable inputs to the Engineer Team, creating a solid foundation for precise, informed structural engineering decisions.

## C.2 ENGINEER TEAM

The Engineer Team systematically conducts engineering evaluations informed by data provided by the Analyst Team. Consisting of specialized agents that focus on structural analysis, structural design, and structural adequacy verification, the team performs iterative analyses through tool utilization and structured data exchanges, evaluating structural integrity and identifying potential risks.

- **Design Engineer**: This agent calculates the structural capacities of individual elements such as beams, columns, and braces, a critical step for ensuring the structural safety and efficacy of proposed designs. Equipped with predefined Python scripts, the agent returns structural capacities and associated section properties to the system memory. These results underpin the selection of optimal yet safe combinations of structural members in comprehensive structural designs.

- **Model Engineer**: This agent synthesizes structured data from the Analyst Team, specifically load information from the Dynamic Analyst and structural model parameters from the Structural Analyst, to execute finite element analyses. Utilizing `OpenSeesPy` and `Opsvis` packages, this agent autonomously generates and runs Python-based finite element analysis, subsequently visualizing results like internal force distributions and deformation shapes for user review. It also engages dedicated Python scripts for automated post-processing, adeptly handling diverse load scenarios beyond seismic loads, such as live loads, thus ensuring automated structural analysis workflows.

- **Verification Engineer**: This agent evaluates structured results provided by the Design Engineer and Model Engineer, deploying Python scripts to systematically verify all critical structural behaviors including tension, compression, bending moments, torsion, deflection, and rotation. Should any structural element fail to meet established performance criteria, the Verification Engineer explicitly identifies the deficiency (e.g., inadequate beam strength at a specific floor), clearly documenting the structural inadequacies and underlying reasons.

Through this structured, iterative evaluation process, the Engineer Team comprehensively captures essential structural and model data. Their meticulous analysis facilitates precise identification of structural responses under specific load conditions and the structural capacity of each element, thereby providing robust support for informed decision-making by the Management Team.

## C.3 MANAGEMENT TEAM

The **Management Team** oversees the operation and coordination of the entire multi-agent system, executing definitive engineering decisions guided by detailed analyses from the Analyst Team and the

specialized insights provided by the Engineer Team. They critically evaluate the integrated findings, examining both quantitative outputs and intermediate analytical processes, to make authoritative and conclusive engineering judgments.

The primary responsibilities of the MASSE Management Team include:

- Coordinating the structural analysis workflow through problem decomposition and task distribution.

- Managing data integration and intermediate results processing, ensuring project timelines and quality standards are maintained.

- Conducting final structural safety evaluations and authoritative decision-making based on comprehensive analytical outcomes.

- Providing standardized safety conclusions and managing system termination protocols.

The Management Team imitates the roles of managers and administrative coordinators in consulting engineering firms, illustrating how essential managerial and coordination functions can be systematically modeled within agent-based workflows.

## D    SPECIFIC CASE TRAJECTORIES

We further present a detailed trace of a representative case trajectory, which illustrates how the multi-agent system progresses through each step of problem-solving. Such traces not only enable us to analyze and debug the system's behavior but also provide valuable insights into potential avenues for optimization and the range of problems the framework can address. By documenting the complete trajectory, the outcomes of the system become highly interpretable. Moreover, presenting key intermediate steps for verification by human engineers significantly enhances the trustworthiness of the system in real-world deployment. An example case trajectory is shown below.

---

**MASSE Analysis Log — Initialization**

**Step 0: System Initialization**

**Log Start:** ========== MASSE ANALYSIS LOG START ==========

**Logger:** Initialized — all terminal output will be saved to log file.

**Database:** - Loading existing vector database from [REDACTED PATH] - RAG seismic database validation successful

**Agent Registration:** 15 functions successfully registered to agents and user_proxy.

**Problem Setup:** Location: Nanaimo, BC System: 2 bays, 8.0 ft beam length, 3 pallets per beam. Beam: 4 in Z-section. Columns: Steel U-channels (3.079 in × 2.795 in × 0.0787 in, 16.0 ft). Braces: Steel U-channels (1.0 in × 1.0 in × 0.054 in, length 4.3 ft). Geometry: 2 beam-columns from $(0,0) \rightarrow (0,16.0)$ and $(3.5,0) \rightarrow (3.5,16.0)$. Bracing: 8 pin-ended trusses connecting specified nodes. Supports: Fixed at $(0,0)$ and $(3.5,0)$. Loads: Applied to left column at 4.0 ft, 8.5 ft, 13.0 ft. Units: Coordinates in ft, forces in kip, stiffness in $kip/in^2$.

**Runtime Config:** - AutoGen version: 0.9.2 - LLM Config: gpt-4o, Temp = 0, Max tokens = 2000 - Runtime logging started (UUID [REDACTED]) - Database path [REDACTED PATH]

**Agents Initialized:** ProjectManager, DesignEngineer, LoadingAnalyst, SeismicAnalyst, DynamicAnalyst, ModelEngineer, StructuralAnalyst, VerificationEngineer, SafetyManager

**System Verification:** Completed — all components verified.

**Log End:** System Initialization finished, structural analysis can proceed.

---

---

**ProjectManager — Step 1: Problem Decomposition and Setup**

**Step 1: Problem Decomposition and Setup**

**User Instruction:** - Call `split_problem_description()` with the full problem text. - After splitting, call `adjust_pallet_weights()` using `number_of_bays` and `number_of_pallets` from memory.

**Tool Call 1:** `split_problem_description` *Input:* Problem description text. *Output:* Problem split into SAA, SDA, LA inputs. Bays = 2, Pallets = 3.

**Tool Call 2:** `adjust_pallet_weights` *Input:* LA_input, num_bays = 2, num_pallets = 3. *Process:* - Loads before update: None - Loads after update: [1875, 1125, 750] (lbs) *Output:* Pallet weights adjusted successfully.

**Result Summary:** - Problem description split into SAA, SDA, LA inputs. - Bays = 2, Pallets = 3. - Pallet weights updated to 1875, 1125, 750 lbs.

**Follow-up Tool Calls:** - `update_saa_input(section_data, load_data)` → SAA input updated successfully, brace coordinates preserved. - `save_analysis_results(filepath="analysis_results.json")` → Results saved.

---

**DesignEngineer — Step 2: Section Design**

**Step 2: Section Design**
**User Instruction:** Extract section info from SDA_input and calculate capacities.
**Tool Call 1:** `extract_section_info` *Input:* Description text from SDA_input. *Output:* Section info extracted — column, brace, beam.
**Tool Call 2:** `calculate_section_capacities` *Input:* Section info (from previous step). *Output:* Section capacities calculated successfully.
**Result Summary:** - Section info extracted from SDA_input. - Element capacities computed and stored.

---

**LoadingAnalyst — Step 3: Building Information**

**Step 3: Building Information**
**User Instruction:** Extract building info from LA_input.
**Tool Call:** `extract_building_info` *Input:* Description text from LA_input. *Output:* Building info extracted — racking system, 3 floors, 3 load points.
**Result Summary:** - Building type: Racking system - Number of floors: 3 - Number of load points: 3

---

**SeismicAnalyst — Step 4: Seismic Parameters**

**Step 4: Seismic Parameters**
**User Instruction:** Get seismic parameters for Nanaimo, BC based on location from building_info.
**Tool Call:** `get_seismic_parameters` *Input:* Location = "Nanaimo, BC". *Process:* Retrieved data using RAG search; 6 relevant document segments found. *Output:* Seismic parameters successfully extracted.
**Result Summary:** - Location: Nanaimo, BC - Sa_0.2 = 1.02 - PGA = 0.446

---

**DynamicAnalyst — Step 5: Load Calculations**

**Step 5: Load Calculations**
**User Instruction:** Calculate seismic loads using data from memory.
**Tool Call 1:** `get_memory_summary` *Output:* Memory contains 14 non-null entries including SAA_input, SDA_input, LA_input, building_info, seismic_parameters, floor_elevations_ft, loads_lbs, section_data.
**Tool Call 2:** `get_memory_data("floor_elevations_ft")` *Output:* [4.0, 8.5, 13.0]
**Tool Call 3:** `get_memory_data("loads_lbs")` *Output:* [1875, 1125, 750]

**Tool Call 4:** `get_memory_data("seismic_parameters")` *Output:* {Sa_02: 1.02, Sa_05: 0.942, Sa_10: 0.037, Sa_20: 0.328, PGA: 0.446, PGV: 0.684}
**Tool Call 5:** `calculate_seismic_loads` *Input:* Floor elevations = [4.0, 8.5, 13.0], Loads = [1875, 1125, 750] *Output:* Seismic loads calculated: F1 = 0.395 kip, F2 = 0.504 kip, F3 = 0.514 kip
**Result Summary:** - Floor elevations and pallet weights retrieved. - Seismic parameters obtained from memory. - Final seismic loads: F1 = 0.395 kip, F2 = 0.504 kip, F3 = 0.514 kip.

---

**ProjectManager — Step 6: Update SAA Input**

**Step 6: Update SAA Input**
**User Instruction:** Call `update_saa_input()` to merge SAA, section, and load data from memory.
**Tool Call 1:** `get_memory_data("saa_input")` *Output:* ...
**Tool Call 2:** `get_memory_data("section_data")` *Output:* JSON containing section properties and capacities for posts, braces, and beams.
**Tool Call 3:** `get_memory_data("load_data")` *Output:* JSON containing load cases: - Seismic: F1 = 0.395, F2 = 0.504, F3 = 0.514 - Live: F1 = 1.875, F2 = 1.125, F3 = 0.75
**Tool Call 4:** `update_saa_input(section_data, load_data)` *Process:* - Original SAA_input length: 365 chars - Brace coordinates preserved - Updated SAA_input length: 365 chars *Output:* SAA input updated successfully with section and load data.
**Result Summary:** - Section and load data retrieved from memory. - SAA input merged successfully. - Structural brace coordinates preserved.

---

**StructuralAnalyst — Step 7: Generate Structural Model**

**Step 7: Generate Structural Model**
**User Instruction:** Call `generate_structural_model()` using updated SAA input.
**Tool Call 1:** `get_memory_data("SAA_input_update")` *Output:* Structural analysis description including: - Two elastic beam-columns from $(0,0)\to(0,16.0)$ and $(3.5,0)\to(3.5,16.0)$ - Fixed supports at $(0,0)$ and $(3.5,0)$ - 8 pin-ended truss braces with specified coordinates
**Tool Call 2:** `generate_structural_model` *Process:* - Found 8 brace coordinates in description - Floor elevations retrieved from memory: [4.0, 8.5, 13.0] - Required load nodes: $(0,4.0)$, $(0,8.5)$, $(0,13.0)$ - Generated 15 nodes total - Generated 8 truss elements + 2 beam-column elements - Model coordinate ranges: X = 0–3.5, Y = 0–16.0 - Applied loads: F1 = 0.395 kip, F2 = 0.504 kip, F3 = 0.514 kip *Output:* Structural model generated with 15 nodes and 10 elements.
**Result Summary:** - Structural model created from SAA input. - Geometry, supports, and braces defined correctly. - Seismic loads applied to correct nodes. - Final model: 15 nodes, 10 elements.

---

**ModelEngineer — Step 8: OpenSees Analysis**

**Step 8: Complete Structural Analysis (OpenSees)**
**User Instruction:** Run `run_complete_opensees_analysis()` using the structural model.
**Tool Call 1:** `get_memory_summary` *Output:* Memory contains 16 non-null entries, including structural_model, load_data, and section_data.
**Tool Call 2:** `get_memory_data("structural_model")` *Output:* JSON model with: - Units: ft, kip, kip/in$^2$ - Materials: E = 29000 - Sections: column (A = 0.705, I = 1.144), brace (A = 0.162) - 15 nodes, 10 elements (2 elasticBeamColumn, 8 truss) - Supports: fixed at base nodes - Loads: seismic forces at nodes $(0,4.0)$, $(0,8.5)$, $(0,13.0)$
**Tool Call 3:** `run_complete_opensees_analysis(model_data)` *Process:* - Generated OpenSees script `structural_model_exec.py` - Analysis executed successfully - Internal forces saved to `internal_forces.json` - Applied load combinations: seismic = 3.340, live = 5.625, envelope = 5.625 *Output:* Internal forces extracted and stored.
**Result Summary:** - OpenSees analysis completed successfully. - Beams: Max tension = 3.34 kip, Max compression = 5.62 kip, Max moment = 7.72 kip·in - Trusses: Max tension = 1.35 kip, Max compression = 1.73 kip - Results saved in memory.

---

**VerificationEngineer — Step 9: Structural Verification**

**Step 9: Structural Verification**
**User Instruction:** Verify structural safety using all analysis data.
**Tool Call 1:** `get_analysis_context` *Output:* Context includes: - Seismic parameters ($Sa\_02$ = 1.02, $Sa\_05$ = 0.942, $Sa\_10$ = 0.037, $Sa\_20$ = 0.328, PGA = 0.446, PGV = 0.684) - Structural model JSON (geometry, sections, loads) - Processed forces: • Beams — max tension = 3.34 kip, max compression = 5.625 kip, max moment = 7.72 kip·in • Trusses — max tension = 1.35 kip, max compression = 1.73 kip - Adjusted SAA, SDA, and LA inputs
**Tool Call 2:** `verify_structural_safety` *Input:* - Capacities: beams and trusses from section design - Demands: seismic loads from nodes *Output:* Safety verification complete — PASS
**Result Summary:** - Verification executed using capacities and demands. - All elements satisfy safety requirements. - Structure passes safety verification.

---

**SafetyManager — Step 10: Final Safety Assessment**

**Step 10: Final Safety Assessment**
**User Instruction:** Use `get_analysis_context()` to obtain all data and provide the final assessment: "FINAL RESULT: STRUCTURALLY ADEQUATE" or "FINAL RESULT: STRUCTURALLY INADEQUATE".
**Tool Call 1:** `get_analysis_context` *Output:* Context includes: - Seismic parameters ($Sa\_02$ = 1.02, $Sa\_05$ = 0.942, $Sa\_10$ = 0.037, $Sa\_20$ = 0.328, PGA = 0.446, PGV = 0.684) - Structural model JSON with geometry, materials, sections, and loads - Processed forces: • Beams — max tension = 3.34 kip, max compression = 5.62 kip, max moment = 7.72 kip·in • Trusses — max tension = 1.35 kip, max compression = 1.73 kip - Safety evaluation: PASS • Post checks — tension ratio = 0.13, compression ratio = 0.28 • Brace checks — tension ratio = 0.18, compression ratio = 0.34
**Final Assessment:** `FINAL RESULT: STRUCTURALLY ADEQUATE`
**Result Summary:** - Full system context retrieved. - SafetyManager confirmed PASS. - Structure deemed structurally adequate. - MASSE analysis log closed successfully.

---

# E EXPERIMENT DETAILS

## E.1 SIMULATION SETUP

We selected a racking system design (Castiglioni et al., 2016) to exemplify typical structural engineering tasks. Racking systems, extensively utilized in warehouses, optimize vertical cargo storage while ensuring structural safety. Structural engineers regularly determine allowable loads and issue certified reports to ensure operational compliance. This scenario effectively integrates core structural engineering tasks—structural analysis, design, and load transformation—aligned directly with the functionality of our proposed agents. Figure 12 illustrates the system configuration. The racking elevation shows a three-tier arrangement with specific load capacities of 1000 lbs (P1), 1250 lbs (P2), and 1750 lbs (P3), while the racking side elevation highlights diagonal bracing employed to achieve lateral stability.

We developed a comprehensive dataset consisting of one hundred distinct scenarios validated through expert-derived ground-truth solutions, facilitating robust evaluation reflective of real-world engineering challenges. Although unique in nature, particularly because loading reports for racking systems are mandated and must be carefully certified by structural engineers specializing in earthquake-prone regions such as British Columbia and California, the racking system scenario is representative of broader structural engineering tasks. The various components of this scenario—such as parameter retrieval, load transformation calculation, structural modeling, structural analysis, structural design, section property determination, section capacity calculation, and structural adequacy verification—are fundamental to structural engineering practice. Therefore, while the agents developed in this framework specifically address the racking system scenario, their design enables easy adaptation to other structural engineering contexts. With minimal modifications, these agents can be employed in alternative multi-agent systems, making them effectively plug-and-play and broadly applicable within the structural engineering domain.

## E.2 EVALUATION METRICS

To accurately simulate the environment of a structural engineering consulting firm, we utilize a dataset derived from real-world racking system projects located in British Columbia, Canada. Our comprehensive dataset enables benchmarking across distinct system components, including agents dedicated to structural analysis, agents focused on structural design, agents responsible for load transformation, and the agents within the whole system that execute the complete structural engineering task. Each component involves multiple agents collaboratively addressing distinct tasks. The dataset encompasses:

**Structural Analysis Agent Benchmark (SAAB):** Evaluates the capability of the structural analysis agents to accurately construct finite element models—including geometry, supports, and load conditions—using OpenSeesPy, based solely on natural language input and internal memory states. The assessment further examines the successful execution of finite element analyses and the accurate retrieval of specified analysis results from OpenSeesPy outputs.

**Structural Design Agent Benchmark (SDAB):** Assesses structural design agents' abilities to accurately interpret dimensional information from natural language inputs, invoke appropriate computational tools to determine necessary properties and capacities, store results correctly, and transfer them effectively into memory states.

**Loading Agent Benchmark (LAB):** Evaluates the proficiency of loading agents in accurately extracting relevant information from natural language inputs, effectively performing Retrieval-Augmented Generation (RAG) operations on supporting documents, and invoking appropriate tools to correctly determine the required applied loads.

**Multi-Agent Structural Engineering Benchmark (MASEB):**

Provides a comprehensive scoring framework designed to objectively assess system performance based on different LLMs.

### E.2.1 EVALUATION JUDGE INSTRUCTIONS AND BENCHMARK RUBRICS

We employ GPT-5 as a LLM Judge to evaluate MASSE's outputs against expert-verified ground-truth solutions. The system instruction is defined as follows.

---

**System Instruction**

System Instruction: MASSE Evaluation Judge (Refined Scheme)
You are an LLM Judge. Your role is to evaluate the performance of the MASSE multi-agent system by analyzing one complete analysis log. You must assign 0–100 points for each benchmark: Structural Analysis Agent Benchmark (SAAB), Structural Design Agent Benchmark (SDAB), Loading Agent Benchmark (LAB), and Multi-Agent Structural Engineering Benchmark (MASEB). Scoring must follow the detailed rubrics below. Your final output must be only a JSON object with four scores, plus total token usage and total time.
The four benchmarks are defined as follows. Structural Analysis Agent Benchmark (SAAB) consists of four components: Model Geometry Accuracy (30 pts), Integration of Section and Load Data (20 pts), OpenSees Analysis Execution (30 pts), and Result Retrieval Accuracy (20 pts). Structural Design Agent Benchmark (SDAB) is divided into Extraction Accuracy (30 pts), Capacity Computation (30 pts), Data Storage and Memory Update (20 pts), and Transfer and Availability (20 pts). Loading Agent Benchmark (LAB) covers Load Extraction (25 pts), Adjustment and Normalization (25 pts), RAG Seismic Retrieval (25 pts), and Load Calculation (25 pts). Finally, the Multi-Agent Structural Engineering Benchmark (MASEB) includes Pipeline Completion (30 pts), Consistency Across Agents (30 pts), Final Result Accuracy (20 pts), and Efficiency and Robustness (20 pts).

---

Figure 11: LLM Judge System Instruction

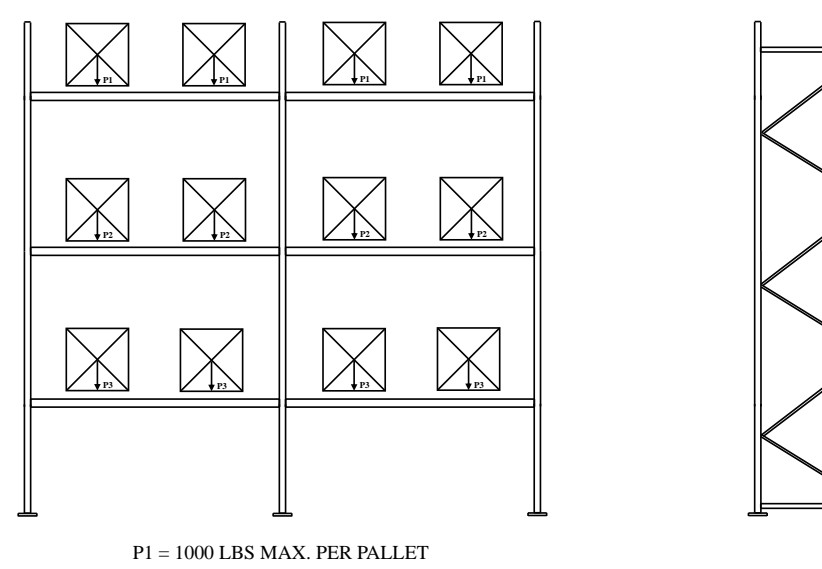

P1 = 1000 LBS MAX. PER PALLET
P2 = 1250 LBS MAX. PER PALLET
P3 = 1750 LBS MAX. PER PALLET

**RACKING ELEVATION**                    **RACKING SIDE ELEVATION**

Figure 12: Racking system example.

# F    PROBLEM EXAMPLE

---

**Problem Description (for structural scheme see Figure 12)**

A steel racking system frame located in Nanaimo, BC is modeled in side elevation as a 2D frame with elastic members and pin-ended truss braces. Coordinates are given in feet (1 ft = 12 in); forces are in kip; stiffness is in kip/in². The overall layout consists of two longitudinal bays in plan, each beam length being 8.0 ft, with a side elevation frame width of 3.5 ft between posts and a post height of 16.0 ft. Beam elevations are placed at 4.0 ft, 8.5 ft, and 13.0 ft, and each beam carries two pallets. The beams are 4 in Z-sections of steel, the columns are steel U-channels 3.079 in × 2.795 in × 0.0787 in with a height of 16.0 ft, modeled as elastic beam–columns with $E = 29{,}000$ kip/in², and the braces are steel U-channels 1.0 in × 1.0 in × 0.054 in, treated as truss elements (pin–pin) with a typical diagonal length of approximately 4.3 ft. The column centerlines are defined from $(0,0)$ to $(0, 16.0)$ and from $(3.5, 0)$ to $(3.5, 16.0)$. The diagonal truss braces connect successive nodes in the following sequence: $(0, 0.5) \rightarrow (3.5, 0.5)$, $(3.5, 0.5) \rightarrow (0, 3)$, $(0, 3) \rightarrow (3.5, 5.5)$, $(3.5, 5.5) \rightarrow (0, 8)$, $(0, 8) \rightarrow (3.5, 10.5)$, $(3.5, 10.5) \rightarrow (0, 13)$, $(0, 13) \rightarrow (3.5, 15.5)$, and $(3.5, 15.5) \rightarrow (0, 15.5)$. The supports are fixed bases located at $(0, 0)$ and $(3.5, 0)$. The weight on each pallet: $P_{4.0\text{ ft}} = 1.75$ kip (1750 lb), $P_{8.5\text{ ft}} = 1.25$ kip (1250 lb), and $P_{13.0\text{ ft}} = 1.00$ kip (1000 lb). Under this configuration, is the structural system safe in this scenario?

---

Figure 13: A racking system problem description

