# OpenReview forum: "Automating Structural Engineering Workflows with Large Language Model Agents"
_ICLR.cc/2026/Conference — ICLR 2026 Conference Withdrawn Submission_

### Official Review · Reviewer_W7cr · 2025-10-28

**Soundness:** 3
**Presentation:** 2
**Contribution:** 2
**Rating:** 4
**Confidence:** 4

**Summary:**

This paper introduces MASSE, a multi-agent LLM system designed to automate complex structural engineering workflows. It addresses the observed failure of single-agent LLMs on long-horizon, multi-tool tasks by decomposing the problem. MASSE mimics a real-world engineering firm, structuring agents into "Analyst," "Engineer," and "Management" teams to handle data extraction, FEM analysis, and final safety verification.

**Strengths:**

1. The paper tackles a high-value problem in a traditionally inefficient industry. The 98% reduction in expert workload is a massive and compelling result that clearly demonstrates the system's practical value.
2. The multi-agent architecture, which mirrors a real-world organizational structure (Analyst, Engineer, Manager), is an intelligent solution for decomposing a complex task that single agents fail to solve.

**Weaknesses:**

1. The entire system and its impressive results are demonstrated on a single, specific task: "racking system design". Claims that the framework is "plug-and-play"  for other, more complex structural engineering tasks are unsubstantiated.
2. The system appears highly reliant on a pre-defined set of tools and specific Python scripts. This "brittleness" may mean it cannot adapt to new problems or different engineering software without significant manual re-engineering.
3. The LLM agents often seem to act as "controllers" that primarily call pre-defined functions and pass structured JSON data. This raises questions about how much novel engineering reasoning is being done by the AI versus how much it is simply executing a very well-defined, pre-scripted workflow.

**Questions:**

How would the system's runtime and token consumption scale to a more complex, 3D structural problem with hundreds of elements and multiple load combinations, as opposed to the 2D frame analyzed?

---

### Official Review · Reviewer_VVfQ · 2025-10-29

**Soundness:** 3
**Presentation:** 3
**Contribution:** 3
**Rating:** 4
**Confidence:** 4

**Summary:**

This paper introduces MASSE (Multi-Agent System for Structural Engineering), a training-free LLM-based framework designed to automate structural engineering workflows. The system employs specialized agents organized into three teams (Analyst, Engineer, and Management) that coordinate to perform tasks including code interpretation, load calculations, FEM analysis, and structural verification. The authors evaluate MASSE on 100 racking system design cases and report 98% time reduction compared to human engineers (from ~2 hours to ~2 minutes).

**Strengths:**

1. Structural engineering is a high-stakes, traditionally manual field where automation could deliver substantial societal benefit. The motivation for applying AI to this domain is compelling.
2. The three-team architecture (Analyst, Engineer, Management) with 9+ specialized agents demonstrates thoughtful decomposition of complex engineering workflows into manageable subtasks.
3. The human evaluation with 11 experienced engineers provides meaningful evidence of practical utility, showing dramatic efficiency improvements.
4. The focus on structured communication (JSON), logging, and explainability aligns well with engineering practice requirements for verification and accountability.
5. Integration of OpenSeesPy for FEM analysis and RAG for building code retrieval shows proper engagement with domain-specific requirements.

**Weaknesses:**

1. Evaluation focuses exclusively on racking system design. Claims about broader applicability to structural engineering (bridges, buildings, etc.) are unsupported.
3. Real structural engineering involves many tasks not demonstrated: irregular geometries, nonlinear analysis, soil-structure interaction, construction sequencing, cost optimization.
3. 100 cases from one geographic region (British Columbia) with reorganized/anonymized data limits reproducibility and generalization assessment.
4. Commercial structural engineering software (SAP2000, ETABS). Other AutoML or AI-assisted engineering tools. Simpler single-agent approaches with stronger models (o1-preview, Claude Sonnet 4)
5. Verification gaps: The "Verification Engineer" agent checks only basic limit states. Real engineering requires: 1) Progressive collapse analysis. 2) Connection design. 3) Fatigue assessment. 4) Serviceability checks beyond what's mentioned.
6. Temperature settings vary across models (0 for some, 1 for o4-mini) without justification
Agent communication rounds limited to 4 without exploration of optimal values
No ablation on number of agents or team composition
7. Figure 1 analogy oversimplifies the human-AI collaboration complexity. Figure 3 is dense and difficult to parse quickly. Appendices are extensive (26 pages) but critical details scattered throughout.
8. JSON communication protocol (Section 4.2): While structured formats help, the paper doesn't address: 1) Schema validation failures. 2) Version compatibility across agents. 4) Recovery from malformed JSON.
9. The paper mentions other systems use "expensive large models" but doesn't compare against these directly on the same tasks. Failure mode analysis: Figure 2 shows single-agent failures (formatting 20%, dependency 50%, logic 30%) but: 1) lacks analysis of multi-agent failure modes. 2) lacks discussion of cascading errors. 3) lacks quantification of error propagation.

**Questions:**

1. How does MASSE handle non-standard structural systems that don't fit the predefined agent workflows?
2. What happens when the Safety Manager declares a structure inadequate? Is there an iterative redesign capability?
3. Can you provide evidence that GPT-5's evaluation judgments align with human expert assessments?
4. What is the false negative rate for safety assessments on a validation set with known failures?
5. How would MASSE extend to 3D structural analysis, nonlinear behavior, or composite structures?

---

### Official Review · Reviewer_3mAS · 2025-10-31

**Soundness:** 2
**Presentation:** 2
**Contribution:** 2
**Rating:** 2
**Confidence:** 3

**Summary:**

This paper introduces MASSE, a Multi-Agent System for Structural Engineering, designed to automate real-world engineering workflows using LLM-based agents. The system mimics professional practice by assigning specialized roles (Analyst, Engineer, Manager) and coordinating their collaboration through structured communication protocols (JSON I/O, shared memory). MASSE integrates finite element (FEM) tools, building codes, and reasoning LLMs like GPT-4o and o4-mini.

The authors evaluate MASSE on realistic racking-system design tasks, benchmarking across four custom datasets (SAAB, SDAB, LAB, MASEB). An ablation study confirms the benefit of structured memory and JSON-based I/O. Overall, MASSE demonstrates that multi-agent LLM frameworks can automate tool-driven, safety-critical workflows in structural engineering without additional training.

**Strengths:**

MASSE represents the first systematic attempt to model structural-engineering workflows through role-specialized LLM agents. Prior work focuses on isolated subtasks (e.g., load calculation, text-to-code), whereas MASSE unifies the full workflow from data extraction to final safety verification.

The system design is clear and well-engineered, featuring persistent structured memory, JSON schemas for inter-agent communication, and integration with FEM solvers. The ablation study convincingly shows why these components matter.

**Weaknesses:**

No quantitative validation: MASSE’s outputs aren’t compared to ground-truth engineering calculations; accuracy claims lack numerical error metrics.

Benchmarks are internal: SAAB/SDAB/LAB/MASEB are self-defined and not externally verified, limiting reproducibility.

Narrow scope: Experiments focus only on racking systems; generalization to broader structural types is untested.

Missing failure analysis: The paper reports only successful runs; reliability or error-rate data are absent.

Weak human study: The 98% time-saving result measures speed, not correctness or safety, with no control group.


Ablation limited: Only memory and JSON are tested; other design choices (role hierarchy, safety logic) remain unexplained.

**Questions:**

How does MASSE handle ambiguous or contradictory information in building codes (e.g., conflicting load provisions across regions)?

Did you test MASSE on non-racking tasks such as reinforced-concrete beam design to assess generality?

How are errors detected and recovered when agents produce inconsistent JSON outputs?

What mechanisms prevent “hallucinated” code or unsafe load assumptions when no ground truth is available?

---

### Official Review · Reviewer_EQx6 · 2025-11-01

**Soundness:** 3
**Presentation:** 3
**Contribution:** 3
**Rating:** 6
**Confidence:** 4

**Summary:**

This paper introduces MASSE, a pioneering multi-agent system that automates structural engineering workflows by leveraging large language models (LLMs). The authors highlight that traditional structural engineering processes are manual, inefficient, and resistant to digital transformation, motivating the need for automation. MASSE addresses this by orchestrating specialized LLM agents into analyst, engineer, and management teams, each handling data extraction, modeling, analysis, and decision validation via structured communication protocols. The system is evaluated using newly proposed benchmarks and a real-world racking system dataset, showing that MASSE dramatically reduces expert workload and operating time by about 98% while maintaining or improving reliability and accuracy compared to human engineers and other LLM configurations. Detailed ablation studies demonstrate the critical impact of memory and structured communication on performance, and runtime analysis reveals a trade-off between communication rounds and efficiency. The discussion emphasizes MASSE’s transparency, safety integration, and potential for real-world impact by reallocating human expertise to higher-level oversight, and concludes that such agent-based automation could reshape engineering and related industries. Future directions include adapting MASSE for practical deployment, refining agent specializations, and enabling real-time feedback for self-improvement.

**Strengths:**

- MASSE is, to the best of current knowledge, the first comprehensive LLM-based multi-agent orchestration for structural engineering. The design closely mirrors real-world engineering team structures, introducing a novel mapping of professional roles to AI agents, and uses structured communication (JSON formats) and memory for stable, auditable automation.

- The empirical validation is thorough: the system is benchmarked against key structural engineering tasks with domain-specific metrics. Performance, efficiency, and scalability (runtime, cost) are analyzed, and ablation studies robustly demonstrate the value of system design choices (agent memory, structured I/O). The human expert comparison demonstrates a 98%+ reduction in manual effort.

- The paper’s problem motivation and potential impact are well articulated. The workflow mapping, modular agent roles, and methodological pipeline are described with clarity, aided by diagrams. The structure targets both AI and engineering audiences.

- Given the global economic and safety relevance of structural engineering, demonstrating that LLM-based agents can match or exceed human reliability on real tasks is an important step for trust in AI in high-stakes domains. The modular architecture and benchmarks may enable rapid transfer to other engineering or procedural professions.

**Weaknesses:**

- Major implementation and experimental details are often relegated to appendices (e.g., agent instructions, simulation setups, dataset specification), making full independent assessment and reproduction difficult from the main text alone.

- It is not clearly stated if or when the anonymized dataset or code will be released. Without public access or a reproducible evaluation protocol, the community may struggle to verify results or build upon the work.

- While single-agent and multi-agent comparisons are made, broader benchmarks against existing automation systems (either commercial or recent LLM-powered research/industry pipelines) are lacking. The practical superiority of the multi-agent breakdown over optimized single-agent setups is not thoroughly quantified (e.g., via cost/failure mode analyses).

- Little qualitative or quantitative error analysis is presented (failure cases, confusion matrices, or limitation scenarios), and statistical reporting of variance or confidence for metrics is sparse except for runtime.

- The structure and blinding of human comparison tasks are not fully described, making it hard to judge the fairness and rigor of the manual vs. MASSE efficiency claims.

- The claim that MASSE generalizes to other engineering/procedural domains remains speculative, with no experimental demonstration. The paper also does not deeply explore real-world adoption barriers (regulatory, integration with legacy tools, practitioner trust).

**Questions:**

- Can the authors clarify whether the anonymized dataset and code used for MASSE’s evaluation will be available to the community? The paper mentions privacy constraints and anonymization but does not specify plans for public release, which affects reproducibility and future research.

- Beyond privacy constraints, could the authors clarify the nature and distribution of their case dataset (e.g., problem diversity, real vs. synthetic split, and any accessible subset for benchmarking)? Without sufficient visibility into data diversity, coverage, and public access, it will be difficult for others to fairly reproduce, benchmark, or extend your evaluation.

- Can the authors provide more description/comparison to strong single-agent or non-agentic LLM baselines, including any relevant complexity (latency/memory/cost) metrics, and justify the multi-agent breakdown over these plausible alternatives? For fair assessment, it is crucial to compare multi-agent systems to optimized single-agent approaches or prior automation pipelines, reporting key trade-offs in complexity and failure modes.

- Could you clarify whether the new dataset and its corresponding ground-truth solutions will be made publicly available, and if not, what measures are in place to ensure reproducibility? Reproducibility of the experimental results hinges on access to the evaluation data; limited or undisclosed data sets may impede this.

- Can you provide more specifics on the structure of the human evaluation—were the 11 engineers working independently on the same instances, what were the precise tasks, and how was their output objectively compared to MASSE results? Understanding the setup, task uniformity, blinding, and scoring criteria is necessary to assess the fairness and rigor of the human vs. MASSE comparison.

- Could you report more granular statistical analyses on the main benchmarks (e.g., standard deviations, confidence intervals, significance testing), not only for runtime but also for the accuracy/score metrics in Table 1? Reporting the variance and conducting statistical significance checks are critical for assessing the reliability and practical impact of performance differences between methods.

---

### Note · Authors · 2026-01-03

I have read and agree with the venue's withdrawal policy on behalf of myself and my co-authors.